# Wrist Hemiarthroplasty of Irreparable Distal Radius Fracture under Wide-Awake Local Anesthetic and No Tourniquet

**DOI:** 10.3390/life12101624

**Published:** 2022-10-18

**Authors:** Thomas Apard, Melissa Odoemene, Jules Descamps

**Affiliations:** 1Ultrasound Guided Hand Surgery Center, 78000 Versailles, France; 2McGovern Medical School, Houston, TX 77030, USA; 3Department of Orthopedic and Trauma Surgery, Hopital Bichat—Beaujon, Assistance Publique—Hopitaux de Paris, 46 Rue Henri Huchard, 75018 Paris, France

**Keywords:** complex intraarticular fracture, distal radius fracture, osteoporotic fracture, proximal row carpectomy, wrist hemiarthroplasty

## Abstract

Distal radius fractures (DRF) are common in elderly patients and the incidence continues to increase with the aging of the population. For irreparable fractures in the elderly, treatment with a reduction cast leads to unreliable results and frequent complications when treated with an anterior plate. Recent studies on hemiarthroplasty for elderly complex wrist fractures have resulted in good clinical and radiologic outcomes, as well as high satisfaction rates. Incorporating wide-awake local anesthesia and no tourniquet (WALANT) in surgical management is beneficial in DRF plating. This technique has not been performed in wrist hemiarthroplasty for an irreparable fracture in an elderly osteoporotic woman. This article describes the WALANT procedure for wrist hemiarthroplasty in a single case, with a detailed description of the technique.

Distal radius fractures are common in elderly patients and the incidence continues to increase as the population ages. In some patients, osteoporosis and the complexity of the fracture make stable fixation almost impossible. Multiple treatments have been suggested, such as the use of an external fixator in addition to a volar plate or a dorsal bridge plate. J-L Roux [1] suggested treating certain very complex acute fractures by utilizing a wrist hemiarthroplasty, as in complex fractures of the superior and distal humerus in elderly patients. Recent studies of hemiarthroplasty for elderly complex wrist fractures have resulted in good clinical and radiological outcomes, as well as high satisfaction rates [2,3,4] and long-term results [5]. According to Herzberg et al. [4], the indications are still limited as there is no uniform definition of an irreparable fracture but it could be an association of fracture characteristics, such as complex, displaced, distal, and cartilage-involved, with circumferential comminution after CT-scan examination.

Fractures of the distal radius are usually treated under general or regional anesthesia with a tourniquet to minimize bleeding and allow for better visualization of the surgical field [6]. Acute physiologic disturbances induced by anesthesia and surgery can lead to decompensation and complications, such as complex regional pain syndrome, complications of sedation, as well as neurological or cardiological complications in elderly patients with extensive chronic disease. WALANT is an alternative anesthetic technique in the case of DRF [7] but it has never been used in hemiarthroplasty for this indication.

After a 30-min lead time, we used the following recipe of WALANT: 1% lidocaine with epinephrine 1:100,000 (30 mL), 8.4% sodium bicarbonate (40 mL), and 0.9% saline (10 mL). We strictly adhered to the safety limit of 7 mg/kg for lidocaine. One intravenous access was required for the injection of antibiotic prophylaxis (cefazolin 2 g).

In total, 10 mL of subcutaneous local anesthetic was injected with a 27-gauge × 13-mm needle through the radial approach at three sites: 4 mL dorsally, 2 mL on the radial side of the radius, and 4 mL volarly, as recommended by Ahmad et al. [7]. An extra 20 mL of local anesthetic was injected under the dorsal skin for the surgical approach. As it takes an average of 30 min for maximal cutaneous vasoconstriction with epinephrine to occur, the efficacy of anesthesia was tested before incision, mobilizing the wrist without any pain. The surgical procedure began when the numeric pain rating scale (NPRS) was zero. An additional 10 mL of local anesthesia was required in the radial medullary canal during the procedure just before introducing the starter compactor. The problem that can occur with wrist hemiarthroplasty is the height of the prosthesis and the ulnar impingement. The WALANT technique helps in ensuring intraoperative control of distal radio-ulnar stability and ulnar-impingement active motion before closure. As a result, we did not need to perform any ligamentoplasty for the distal radio-ulnar joint stability.


**Case presentation**


An 83-year-old right-handed woman presented with pain, swelling, and deformity of the left wrist after falling on the street. She had landed on her outstretched left hand and was subsequently unable to move her wrist. No wounds or neurovascular injuries were noted. Radiographs and a CT scan of her left wrist showed a complex distal radius fracture (AO classification 23.C3 type; Figure 1A,B). She was taking medications for diabetes, aspirin for arterial hypertension, and Vitamin D for severe vertebral osteoporosis. A hemiarthroplasty (Cobra, Groupe Lepine^®^, Genay, France) under the WALANT procedure was proposed as an outpatient procedure. Aspirin was never discontinued and no adverse effects or complications were noted. No sedative drugs were administered.

The patient was operated on 10 days after the trauma. The wrist was accessed through a 6-centimeter straight dorsal incision involving Lister’s tubercle. The dorsal venous network was carefully preserved. The extensor retinaculum was dissected between the third and fourth compartments. A subcapsular-periosteal detachment was performed to preserve the fourth compartment. After exposure of the joint space, the epiphyseal bone fragments were resected gently with a gouge. This resection was limited to preserve periprosthetic bone material. Reaming of the radial medullary canal was performed in the axis of the third metacarpal, successively increasing the size of the bone reamer. The trial implant was examined intraoperatively by fluoroscopy to verify correct positioning, ensuring appropriate implant height without impingement of the ulnar head. The final implant was not cemented. The final assessment was performed by asking the patient to move her wrist under fluoroscopy (Appendix A). Pain killers were injected intravenously immediately after the procedure. 

The patient was immobilized for one week with an anterior splint, followed by a removable cast extending below the elbow to start physical therapy. Radiological and clinical examinations were performed at 1 week (Figure 1C,D), 1 month, 3 months, and 6 months. After one month, post-operatively, she regained full flexion, pronosupination, and mobilization of the fingers, and pain was at 1 of 10 on the NPRS at rest. However, extension remained at a 50° extension deficit compared to the contralateral side (Appendix A). She was able to perform daily tasks. After three months, the extension deficit was 20°. At 6 months, no complications had occurred and she rated her wrist at 80% of a normal wrist using the Lyon Wrist Score, with a radiograph showing complete integration (Appendix A).

We conclude that wrist arthroplasty using WALANT may be an option to restore function in elderly distal radius fractures.

## Figures and Tables

**Figure 1 life-12-01624-f001:**
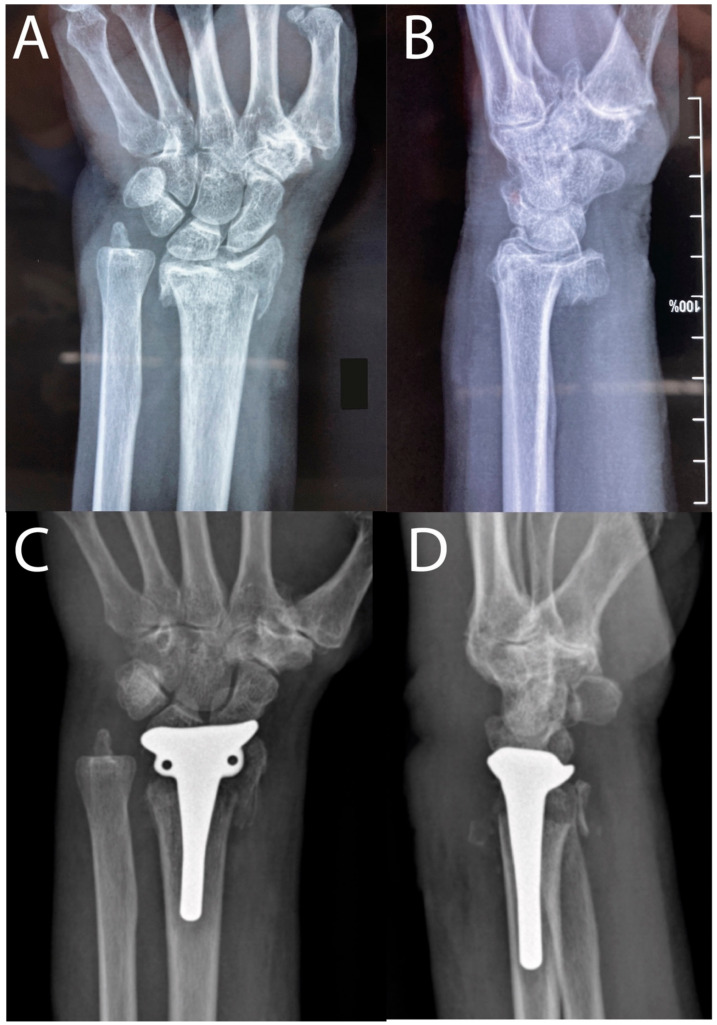
(**A**) Preoperative X-ray AP. (**B**) Preoperative lateral view. (**C**) Postoperative AP. (**D**) Postoperative lateral view.

## Data Availability

All data are included in the manuscript.

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
