# Peer review of "Wrist Hemiarthroplasty of Irreparable Distal Radius Fracture under Wide-Awake Local Anesthetic and No Tourniquet"

_life, 2022, doi:10.3390/life12101624_

Round 1
Reviewer 1 Report
- The main question is to show the ability to do this complex procedure using a type of local anesthesia which may be much safer for the older age patient While the technique of hemi arthroplasty for treatment of complex distal radius fracture in the elderly has been well published, this is the first time it appears in publication using local anesthesia
- It would help if a true lateral follow up xray is provided as the current xray is oblique.
- The paper is well written and easy to read.
- The conclusions are sound, address the issues raised and respond to the questions posed
Reviewer 2 Report
Thank you for the opportunity to review this case report. I have some comments for the authors.
the topic of complec distal radius fractures in older peolpe is interesting, this fracture is quite distal and intra-articular and both a cast or a volar plate could have been good options.
-please elaborate on the indications for hemiarthroplasty.
-what about the bone ulnarly and the DRU-joint. Are the ligaments sutured to the hole in the implants? Is the DRU-joint stable?
-line 86, the pain score, is this at rest or during activity?
